# Response of Macrophyte Diversity in Coastal Lakes to Watershed Land Use and Salinity Gradient

**DOI:** 10.3390/ijerph192416620

**Published:** 2022-12-10

**Authors:** Mirosław Grzybowski, Paweł Burandt, Katarzyna Glińska-Lewczuk, Sylwia Lew, Krystian Obolewski

**Affiliations:** 1Department of Tourism, Recreation and Ecology, University of Warmia and Mazury in Olsztyn, Oczapowskiego Str. 5, 10-719 Olsztyn, Poland; 2Department of Water Management and Climatology, University of Warmia and Mazury in Olsztyn, Łódzki Sq. 2, 10-719 Olsztyn, Poland; 3Department of Microbiology and Mycology, University of Warmia and Mazury in Olsztyn, Oczapowskiego Str. 1a, 10-719 Olsztyn, Poland; 4Department of Hydrobiology, University of Kazimierz Wielki in Bydgoszcz, 85-090 Bydgoszcz, Poland

**Keywords:** macrophyte diversity, coastal lakes, salinity, hydrological connectivity, land use, watershed, phytocenotic indices

## Abstract

Coastal lakes are subject to multiple stressors, among which land use, hydrological connectivity, and salinity have the greatest effect on their biodiversity. We studied the effects that various land cover types (CORINE) of coastal lake watersheds had on macrophyte diversity in ten coastal lakes along the southern Baltic coast as characterised by twelve phytocenotic indices: these being a number of communities, Shannon–Wiener diversity, evenness, and indices of taxonomic distinctiveness of plant communities: vegetation coverage; colonisation index; share of the phytolittoral area in the total lake area, as well as shares of nympheides, pondweeds, charophytes, marine, emerged and submerged communities in the total lake area. The effects were checked for three groups of lakes distinguished by differences in salinity–freshwater (F, 5), transitional (T, 4), and brackish (B, 1)—in which a total of 48 macrophyte communities were identified. The most abundant in aquatic phytocoenoses were lakes of T type. A partial least squares regression model (PLS-R) showed a stronger impact of land-use types in immediate vicinities and entire watersheds than the impact of physico-chemical properties of water on phytocenotic indices in the lakes. Macrophyte diversity was relatively low in urban and agricultural catchments and relatively high in forest and wetland areas. Agriculture had a negative impact on the number of macrophyte communities in F lakes and, in T lakes, on the number of macrophyte communities, biodiversity, evenness, and proportion of emerged, submerged, and marine communities. Urban areas contributed to lower values of evenness, vegetation coverage, and share of marine communities in F, but, in T, to lower the number of macrophyte communities, evenness, and proportion of submerged and marine communities. Our results confirm the significant impact of land use on macrophyte diversity in coastal aquatic ecosystems. Combined analysis of anthropogenic and natural descriptors is a prerequisite for analysing human threats to biodiversity in coastal lakes. Macrophyte community-based measures of biodiversity are sensitive indicators of anthropogenic impact on the ecological condition of coastal ecosystems.

## 1. Introduction

Coastal areas of seas and oceans cover less than 15% of the Earth’s land area but are inhabited by more than 60% of the world’s human population [1]. Coastal lakes are found on all continents and account for 13% of global coastline length. They belong to the most bioproductive and valuable ecosystems but simultaneously to the most threatened habitats [2,3]. In the European Union, coastal water bodies are protected under Natura 2000 network as “Habitats of Community Interest”. The European Union’s Habitats Directive (HD) [4] classifies coastal lakes as a habitat that belongs to “coastal lagoons” (code 1150), whose subtype “coastal lakes” is coded 1150-2. Thus, the environmental objectives for the protection of coastal lake habitats set out in the HD should meet the objectives required by the Water Framework Directive [5].

The impact of both natural (marine and wind erosion) and anthropogenic pressures (climate change, urbanisation, agriculture, tourism, etc.) on coastal habitats was reflected in the EEA report (2021), which stated that, of the 901 coastal water bodies tested in 2021, the status “above good” was found only for 33%, which is a lower proportion than for other types of surface water bodies. Determining the sources of unsatisfactory conditions (“below good”) is quite complex and difficult because they all exhibit heterogeneous characteristics due to (i) large variability in abiotic and biotic parameters of lake basins, (ii) time-variable and spatial impacts of sea and land influences, and (iii) differences in size and land use of their catchments.

The functioning of many aquatic ecosystems along the seacoast is indisputably driven by the intrusion of marine waters. Saline water is often reported as a primary determinant of ecosystem health, highlighting the importance of seawater on the biotic and abiotic elements of these habitats [6]. However, coastal ecosystems are the terminal hydrographic features that water flowing from watersheds must encounter before reaching the oceans. Thus, the lakes are subject to many consequences of the human activities that take place in their watersheds [7]. The impact of land use on downstream aquatic environments through pressures exerted by point-source pollution, non-point-source pollution, and modifications to flow regimes caused by abstractions or regulation and morphological alterations is highlighted. Human activities in watersheds may accelerate or slow the translocation of sediments and pollutants or the accumulation of sediments and pollutants in coastal wetlands, lakes, and estuaries [8].

Among the main causes of coastal lake degradation, agriculture in lake watersheds has a clear impact due to the excessive input of nutrients into the water [9]. Moreover, Cieśliński and Olszewska [10] pointed out that lakes along the southern Baltic coast were often used as wastewater reservoirs for municipal sewage and agricultural and industrial waste, which for years degraded their water quality. Some were modified by hydrotechnical infrastructures (canals, weirs) to allow agricultural and tourism development, energy production, and flood control.

According to WFD requirements, ecological status should be determined for transitional and coastal waters based on indicators of biological quality (phytoplankton, macrophytes, phytobenthos, benthic invertebrate fauna, and fish) and supporting physico-chemical parameters (nutrients, aeration, temperature, transparency, salinity, and river-basin-specific pollutants) as well of indicators of hydromorphological properties. However, some authors [11,12] have found weak relationships between stressors and water quality and biological (benthic) indices, and this is seen as the main weakness of WFD monitoring and evaluation. The weakness of relationships is particularly seen in various types of coastal waters, which are characterised by a diversified internal structure [13] and simultaneous pressures from their watersheds and the sea. Karus and Feldmann [2] have suggested that macrophytes should be used as indicators of the ecological condition/potential of lakes because various phytocoenoses have different environmental requirements. The usefulness of macrophytes as indicators is justified by their long-term response to changes in environmental conditions (which is similar to that of most sessile organisms) [14,15]. Submerged macrophytes are recommended as one of the four biological elements (along with phytoplankton, benthic invertebrates, and fish) to be used to describe ecological quality [5,16]. Methods for characterising macrophytes in lakes at the community level have traditionally focused on the presence of ‘indicator’ species, often based on expert judgement [17,18]. Due to the high variability of lakes and macrophyte parameters, there are few macrophyte-based classification systems for assessing the ecological status of coastal lakes [19,20]. In the case of coastal ecosystems, the concept of macrophyte vegetation succession along environmental gradients of the study area can also be based on Luther’s extensive autecological studies [21,22]. In view of this, changes in taxonomic composition, an abundance of indicator species, and depth of growth of macrophytes are often recommended for use as a set of direct, reliable indicators for assessing the health of coastal ecosystems [2,14]. For this reason, it is justified to search for the relationship between the biotic structure of macrophytes and land use of the watersheds of coastal lakes against the background of marine water supply.

We hypothesised that there is a relationship between the structure of macrophytes and land use in coastal lakes’ watersheds, despite the impact of significant marine interference on the health of coastal ecosystems. The aim of this study was to: (i) characterise the species structure of macrophytes in lakes along the southern Baltic Sea coast, (ii) identify primary stressors of the syntaxonomic and functional diversity of macrophytes in type-specific coastal lakes, and (iii) determine and evaluate the suitability of ten selected macrophyte metrics for the structure of land use in direct and indirect lake catchments and their response to haline stress.

## 2. Materials and Methods

### 2.1. Study Sites

The study was conducted at ten coastal lakes along the southern Baltic coast (Figure 1).

Their water table areas range from 53 ha (Lake Ptasi Raj) to 7044 ha (Lake Łebsko). The lakes are shallow and have an underdeveloped shoreline. The influence of westerly winds, which mix the water to the bottom, gives these water bodies an extremely polymictic character and a tendency to over-fertilisation (secondary pollution). However, they differ in the physico-chemical properties of the water depending on their degree of connection with the sea and the influence of the catchment (Appendix A). The operating principles of Baltic coastal lakes, conditioned by both marine and terrestrial factors, are explained by the theory of alternative stable states in shallow lake ecosystems (brackish vs. freshwater) separated by adaptive cycles (transitional lakes) [6]. The average concentrations of mineral forms of nitrogen and total phosphorus in the lake water are relatively high and amount to 0.58–1.47 mg/L and 0.16–0.87 mg/L, respectively (Appendix A). The most frequently recorded water transparency (measured as Secchi disk depth) does not exceed 0.4 m.

Ecological conditions in the Baltic coastal lakes are determined by hydrological connectivity with the sea. However, the low salinity (<7 PSU) typical of the Baltic Sea waters causes the coastal aquatic ecosystems to be, at most, brackish [24]. Prior to analyses, the lakes were grouped by their average salinity (PSU) based on Euclidean distance and Ward’s agglomeration method (Figure 2A). As a result of the clustering procedure, three groups of coastal lakes were selected: (a) freshwater lakes (F) without direct connection to the sea and with salinity at limnetic level (Dołgie Wielkie, Jamno, Liwia Łuża, Sarbsko, and Wicko Przymorskie); (ii) transitional lakes (T) where regular or occasional seawater intrusions change the salinity and the lake has a limnetic–oligohaline character (Resko Przymorskie, Łebsko, Gardno, and Kopań); (iii) brackish lakes (B) continuously fed with seawater of average salinity, also identified as mesohaline (Ptasi Raj), (Figure 2B).

For each lake, the total and direct watershed areas were determined based on a digital terrain model (DTM) and compared with the hydrological map of Poland (1:50,000). The areas vary widely, ranging from 410 ha for Ptasi Raj to 156.9 × 10^3^ ha for Łebsko, whereas total and direct watersheds of lakes Ptasi Raj and Dołgie Wielkie do not differ. The share of lake area in the total watershed area does not exceed 22% (Dołgie Wielkie), whereas its share in the direct watershed area ranges from 13% for Ptasi Raj to 71% for Jamno, which means that the lakes are in danger of being polluted from land sources to varying degrees.

The Corine Land Cover (CLC) database, created by the EU and coordinated by the European Environment Agency, served as a source of information on land use [23]. The CLC has one of the most detailed land-use classification systems in the world, presenting land-use data at three levels: Level 1 includes the five main classes of land use; Level 2 includes 15 land-use sub-classes; Level 3 includes 44 sub-classes [25]. CLC classes were used in the factor analysis to evaluate the effects of direct and total watershed land uses on the species structure of macrophytes in the lakes studied. The CLC uses a minimum mapping unit of 25 hectares based on visual interpretation of high-resolution satellite imagery.

Among the studied lakes, the entire watershed areas of Resko, Liwia Łuża, Wicko, and Kopań are mainly used for agriculture (Figure 1), while the watersheds of lakes Jamno, Sarbsko, Łebsko, and Gardno are used for forestry and agriculture. Forestry use dominates in the watersheds of lakes Ptasi Raj and Dołgie Wielkie. The highest percentage of anthropogenic use characterises lakes Jamno, Wicko, Resko, and Liwia Łuża, whereas the lowest is found for lakes Dołgie and Ptasi Raj.

### 2.2. Phytocenotic Studies and Evaluation

The study was carried out twice during growing seasons from July to September in 2014 and 2015. Coastal vegetation was studied on selected transects up to 500 m wide, along which macrophyte communities were identified as clusters/aggregations of plants, mostly consisting of one species ≥ 1 m^2^, covering at least 25% of this area. At each lake, surveys were conducted on at least half of the transects resulting from the Jensén formula [26], but much wider than the method assumes, 30 m, which is acceptable according to the method by Ciecierska [27]. The Jensén formula makes it possible to determine, based on water table area and shoreline length, the sufficient number of macrophyte survey transects to assess the status of lakes [27,28]. Due to the scattered nature of the patches and their low cover, quantitative measurements of macrophyte density were not performed.

The values of ten phytocenotic diversity indices were calculated according to the methods proposed by Ciecierska and Kolada [28]. The following indices were determined for each lake: the number of macrophyte communities (S), diversity (H’), evenness (J’), and share of emerged (%Emerg) and submerged vegetation (%Subm), pondweeds (%Pota) and nympheides (%Nymph). The vegetation cover of an entire transect was expressed by colonisation index (Z, %), the share of phytolittoral area in the total lake area (N, %), the share of marine communities (%Mar_com) and charophytes (%Chara), and vegetation coverage (C_max_). The species diversity (H’) was calculated according to Shannon–Wiener and species evenness (J’) after Pielou.

### 2.3. Water-Quality Parameters

Water-quality parameters were used in the present study as explanatory variables having the potential impact on the macrophyte diversity of coastal lakes. Physico-chemical parameters of water were measured seasonally (May, July, and October) over a two-year period at five to seven monitoring sites in each lake. Water parameters were measured with a YSI 6600 multiparameter probe (YSI Inc., Yellow Springs, OH, USA) shortly after the probe was calibrated: water temperature (T, °C), pH, dissolved oxygen (DO, %), Chl-*a* (µg/L), salinity (PSU) and conductivity (EC, µS/cm). In the laboratory, water samples were analysed for N-NO_2_^−^, N-NO_3_^−^, N-NH_4_^+^, TP, P-PO_4_^3−^, total organic carbon (TOC), and dissolved organic carbon (DOC). The details of the sampling strategy have been described by Obolewski et al. [29]. Analytical procedures followed the APHA Standard Methods [30]. The physico-chemical parameters of water in each lake are listed in the supplementary material (Appendix A).

### 2.4. Statistical Analyses

The macrophyte and environmental data collected for each lake were statistically analysed to: (a) characterise and compare macrophyte community composition in ten coastal lakes representing three habitat types that differ in salinity levels; (b) identify phytocoenotic indices for each of the three salinity types of coastal lakes; (c) relate land-use determinants to macrophyte structure in the lakes. We tested the significance of differences between lake types (freshwater, transitional, and brackish) as grouped (explanatory) variables and macrophyte data (dependent variables) using one-way analysis of variance (ANOVA) with Tukey’s multiple comparison test (*p* < 0.05) as a post-hoc procedure. Correlation analysis, mean abundances, and standard deviations (±SD) were calculated using Statistica™ 13.1 (Tibco Inc., Palo Alto, CA, USA, 2021). Prior to statistical analyses, the normal distribution of the biological data and environmental variables was tested using the Kolmogorov–Smirnov test (*p* < 0.05). Data sets were log_10_(x + 1) transformed to stabilise the variance.

To visualise differences in macrophyte community composition in the coastal lakes studied, a heat map was constructed, and two-way cluster analysis (TWCA) was performed in PC-ORD 6.08 (MjM Software Design, UK) using the Sorensen (Bray–Curtis) distance measure and flexible-β at −0.25 as the linkage algorithm. The data were relativised by column.

The response of macrophyte communities to multiple variables (stressors) was analysed using multivariate statistical analyses. Macrophyte data (Y) were analysed against water quality and land use in lake catchments as predictors (X). Partial least squares regression (PLS-R) is a multivariate statistical tool for relating two data matrices by a linear multivariate model [31]. The PLS-R model combines the explanatory variables (X) to create a new set of latent variables (LVs) that capture the variance on X correlated with Y, and it computes a regression vector. The strength of the model was taken from two values: the R2 value and the Root Mean Square Error of Cross Validation (RMSECV) value. Only significant relationships (*p* < 0.05) with the environmental indices revealed in the PLS-R analysis were included in the analyses [32].

To rank individual predictors according to their discriminative potential and for the identification of substantial explanatory variables in PLS regression models, variable identification coefficients (VID) were calculated [33]. VIDs are defined as correlation coefficients between the original X-variables and the Y-variables predicted by the PLS models. X-variables with a contribution of VID values higher than 0.85 (VID ≥ |0.85|) were considered statistically significant.

## 3. Results

### 3.1. Spatial and Qualitative Variation of Macrophyte Communities in Coastal Lakes

During the study, we observed a total of 48 macrophyte communities in the ten coastal lakes along the southern Baltic coast, with the total number of macrophyte taxa per lake varying from eight to 23 (see Appendix A).

The rush phytocenoses *Scirpus* spp. (*S. tabernaemontani*, *S. maritimus*), *Typha* spp. (*T. latifolia*, *T. angustifolia*), *Buttomus umbellatus*, and *Acorus calamus* are of greater diagnostic importance in transitional lakes, while phytocenoses of nympheids (*Nymphaea* sp., *Polygonum amphibium*), rushes (*Scirpus lacustris*, *Phalaris arundinaceae*, *Glyceria maxima*, *Iris pseudoacorus*, *Phragmites australis*, *Carex acutiformis*), and water communities (*Potamogeton perfoliatus*, *Myriophyllum spicatum*) are typical of inland eutrophic ecosystems that have diagnostic significance for freshwater lakes. Under the specific hydrochemical conditions of lake Dołgie Wielkie, characterised by the lowest values of salinity and conductivity [24], communities typical of dystrophic lakes developed, indicating an abundance of humic acids, such as *Carex rostrata*, *Comarum palustre*, *Carex paniculata,* and *Equisetum fluviatile*, accompanied by nympheids. They formed a separate group (see heat map in Figure 3) that, unlike the other lakes studied, did not have submerged vegetation.

Ptasi Raj, on the other hand, is a brackish lake and is distinguished from the other coastal lakes by its predominant submerged vegetation. It is characterised by a considerable number of charophyte beds, which are absent or sparse in the other lakes studied, as shown in a heat map (Figure 3).

Macrophyte species differed in number between lakes. They were most abundant in freshwater lakes, where 35 species were noted. Helophytes (23 species) dominated, while Elodeides and Nympheides were represented by six species each. No charophytes were found in freshwater lakes. In transitional lakes, we found 32 species, of which 50% were Helophytes. Elodeides were represented there by nine species, Nympheides by five species, and Charophytes by just two species. In brackish lake Ptasi Raj, the fewest species (8) were found, among which Charophytes dominated (50%). We found three representative species of Elodeides there and one of Helophytes. As many as six taxa of submerged vegetation were found in freshwater lakes, seven in brackish lakes, and eleven in transitional lakes.

One-way ANOVA analysis showed statistically significant (*p* < 0.001) differences between the three groups of Baltic coastal lakes studied (freshwater F, transitional T, and brackish B) in terms of composition and syntaxonomic diversity of macrophytes (Figure 4).

The average Shannon–Wiener diversity index (H’) was 1.35 and showed no significant differences among the studied lake types. The Pielou index (J’), which measures the evenness of distribution of all communities, averaged 0.45 and differed statistically among the lake types. The studied lakes showed significant lake-type-dependent differences in vegetation cover (C_max_), colonisation index (Z), total submerged vegetation (%Subm), the share of charophytes (%Chara), and the share of marine communities (%Mar_com).

### 3.2. Factors Influencing the Diversity of Macrophytes in Coastal Lakes

To illustrate the multidimensional data structure, biplots of scores and correlation loadings were created based on the PLS-R model. The biplot (Figure 5A) summarises the contribution of environmental factors (explanatory variables, X): total and direct catchment areas, land use, and physico-chemical parameters of water that control macrophyte diversity (dependent variables, Y) in the studied coastal lakes.

The data matrices for coastal lakes are well described by two significant latent variables (LV) in components 1 and 2 (Figure 5A,B). In the first LV, 24.6% of the variance in the X matrix explains 46.8% in the Y matrix. In the second LV, 13.2% of the X matrix explains 11.9% of the Y matrix. Combining the entire model (LV1 and LV2), 58.9% of the variance in the response data was used to explain 46.8% of the Y variables. Q2, as a measure of predictive accuracy, showed that Component 1 (Axis 1) explaining macrophyte diversity (Q2 = 0.467) was significant (Q2 threshold > 0.097, corresponding to *p* < 0.05) (Figure 5C).

Vegetation in Baltic coastal lakes, as characterised by phytocenotic indices, varies in terms of species composition and cover, which depend on the characteristics of land use in their total and direct catchments rather than on the physico-chemical parameters of the lake water (Figure 5A). VIP diagrams (Figure 5D) show the relative importance of predictors for both components 1 and 2. Among VIPs > 0.8, based on Wold’s criteria, we found wetlands (CLC IV) and forested areas (CLC III) of significant importance to the corresponding dependent variables (Y, macrophyte indices). The presence of wetlands, particularly saline (CLC IV), promotes marine communities and charophytes while it limits the number of macrophyte species (S). The increased contribution of forested areas (CLC III) in direct catchments has a significant impact on J’, %Subm, N, Z, and C_max_. Anthropogenic areas (represented by CLC classes I_d_, I_t_, II_t_) contributed to a reduction in the majority of phytocenotic indices, except for the emerged vegetation.

Salinity and water transparency (SD) in coastal lakes supported the development of marine species, including charophyte beds. Salinity alone exhibited geographic differences, being generally higher in lakes disconnected from the sea (Figure 5B, brackish lake Ptasi Raj) than in lakes with occasional connectivity (T). The impact of saline water stimulated other indices, namely: maximum occupation rate (C_max_), the share of phytolittoral area (N), and colonisation index (Z). Other water quality parameters (TP, P-PO_4_, N-NH_4_) corresponded to anthropogenic land use of class I (urban) and II (agricultural). However, the relation was even stronger for the land-use class of total watershed areas. The size of total and direct watersheds did not contribute significantly to the PLS-R model.

The high variability of the land-use predictors required that hydrological connectivity be considered as a qualitative predictor distinguishing the three types of lakes. Thus, the most discriminative predictors of macrophyte diversity with absolute VIDs greater than 0.85 (VID≥|0.85|) for freshwater and transitional coastal lakes are listed in Table 1. Brackish lakes were represented by only one lake (Ptasi Raj) with the domination of marine species and were excluded from the list.

Among land-cover classes that positively correlated with a number of species (S), we identified cultivation areas (CLC 242_t_) and mixed forests (CLC 313_t_) for freshwater lakes and water bodies (CLC 512_d_), agriculture lands with significant areas of natural vegetation (CLC 243_t_) and subclasses of class III (mixed forests (CLC 313_t_), beaches, dunes, sands (CLC 331_d_), transitional woodland-shrub (CLC 324_t_), and broad-leaved forest (CLC 311_t_) for transitional lakes. The PLS-R model showed that the most adverse effects on S in transitional lakes are associated with anthropogenic land uses such as discontinuous urban fabric (CLC 112d, VID = −0.99) and pastures (231_d_, VID = −0.98).

Forests (particularly mixed forest (CLC 313_d_), transitional woodland-shrub (CLC 324_t_) in watersheds of freshwater lakes were related to the following indices: evenness (J’) and biodiversity (H’), the share of phytolittoral area (N), the share of submerged macrophytes (%Subm), and colonisation index (Z), for which VIDs > 0.91. In transitional lakes, J’ was highly impacted by the presence of inland marshes (CLC 411_d_; VID = 0.99) and mixed forests (CLC 313_d_; VID = 0.98), while H’ was positively related to industrial or commercial units (CLC 121), beaches, dunes, sands (CLC 331) and peat bogs (CLC 412), all with VIDs = 0.93. The CLCs contributing to H’ decline in transitional coastal lakes included non-irrigated arable lands (CLC 211) of both total and direct watersheds with negative VIDs = −0.91 and −0.86, respectively.

Mixed forest (CLC 313) in direct watersheds of both freshwater and transitional lakes promoted maximum occupation rate (C_max_). However, the impact of sparse built-up areas (CLC 112_d_ and 112_t_) limited C_max_ in freshwater coastal lakes. The share of emergent plants in freshwater was adversely correlated with mixed forest (CLC 313_d_) and woodland-shrub areas (CLC 324_t_), with VIDs amounted to −0.92 and −0.91, respectively. In transitional lake catchments, the role of inland marshes (CLC 411_t_; VID = −0.99), cultivation areas (CLC 242_t_; VID = −0.97), and water bodies 512_t_ (VID = −0.97) dominated (Table 1).

## 4. Discussion

The coastal lakes studied along the southern coast of the Baltic Sea are a rather rare example of ecosystems whose ecological status can be assessed by taking into account the different degrees of connection with the sea and the impact of land use in their watersheds. Both of these factors exert significant pressure on aquatic ecosystems and usually contribute to the decline of their ecological values. In the case of coastal lakes, the use of macrophyte metrics or indices is a specific tool that demonstrates pressure-response relationships with land- and sea-based impacts. It was developed to identify threats such as eutrophication and salinity. Pressure-response relationships are often based on proxy indicators such as land use or multi-pressure indices [34,35]. In addition, land use is considered a proxy for other pressures, including pollution and hydromorphological changes [36,37]. Many methods refer to land use because Corine Land Cover data are readily available [38].

Macrophyte indicators, along with phytoplankton, benthic invertebrates, and fish fauna, are among the biological quality elements (BQEs) defined in the European Water Framework Directive [5] and used to assess human-induced multiple pressures that can interact in additive, synergistic, or antagonistic ways [39]. The vegetation metrics used in our study appear to be a suitable diagnostic tool for assessing the ecological status of coastal lakes. However, relationships with all water body types have not yet been described for all BQEs [34]; for example, the relationship between land use and the macrophyte quality component of coastal lakes has not been studied. Many authors agree that the advantage of plants is their long-term response to changes in environmental conditions [40,41]. Therefore, they are suitable for identifying the causes of ecological status degradation and guiding the choice of appropriate management measures [42].

The ten macrophyte measures used to assess land-and seawater-related pressures were tested in our study on nine coastal lake ecosystems under conditions of their limited, seasonal, or permanent seawater recharge. There is hydrological connectivity between a lake and the sea limits; however, the role of the watershed impacts the phytolittoral structure. Our study showed that phytolittoral structure depends to varying degrees on nutrient input and, thus, indirectly on factors in the watershed [43], but also, and very importantly, phytolittoral structure reflects hydrochemical conditions in a given lake [44].

We have demonstrated a positive correlation between the development of submerged aquatic vegetation and features commonly considered to indicate eutrophication caused by anthropogenic use of lake catchments, namely agriculture and settlements. This finding is also supported by other studies, e.g., [16,43]. There is value in restoring submerged macrophyte diversity in the coastal lakes studied, as this will facilitate the creation of a clean water condition in these shallow eutrophic lakes [45]. However, most submerged macrophyte restorations are often unstable and cannot maintain a stable state of clean water, possibly because species and functional diversity have not been fully considered, e.g., [46,47]. Zhou et al. [8] showed that a community of submerged aquatic vegetation consisting of multifunctional, species-rich groups (eight species or more) is conducive to building stable submerged macrophyte communities and achieving a stable state of clean water. In the context of our study on coastal lakes, this suggests the need to restore species-rich communities of submerged aquatic vegetation. Currently, there are no more than seven submerged species in each of the lakes surveyed: the highest number was found in the brackish lake Ptasi Raj (7 species), while no submerged vegetation was found in the lake Dolgie Wielkie, under the influence of humic substances and the dystrophy process.

Our study confirmed a negative correlation between macrophyte diversity indicators (except for the proportion of emergent vegetation) with agricultural use and urbanised areas. Increasing agricultural intensity and urbanisation in lake watersheds leads to increased nutrient delivery to the lake littoral [48,49]. This leads to eutrophication, which ultimately reduces lake macrophyte diversity [50]. Industrial or commercial areas are often associated with increased salinity and may have a greater impact on the distribution of marine species [6,13]. A positive relationship between the proportion of agricultural land in the watershed and increases in lake trophic status has long been recognised, e.g., in North America [51], Europe [52], Asia [53], and New Zealand [54]. However, research on coastal lakes has not been conducted. According to Novikmec et al. [55], the entire watershed of the water body should be considered, as the narrow buffer zone promoted in the literature [34] does not provide sufficient protection.

In transitional lakes, a stronger influence of anthropogenic catchments of urban type on macrophyte diversity indicators, such as the number of plant communities, H’ diversity index, and proportion of submerged vegetation, was observed. Similar relationships were observed in freshwater inland lakes [28], where the proportion of submerged and emerged vegetation depended on the pollutants outflow caused by agricultural and urbanised areas. The proportion of forests in the catchment area favoured an increased number of marine taxa.

Our results are consistent with Mozgawa [56], who found that marshes and wetlands are the most effective natural filters for nutrient input into lakes. Forests, meadows, and pastures are moderate barriers, while cropland and urban areas do not appear to be such barriers. Our results showed that the presence of wetlands and water bodies in coastal watersheds had a strong positive influence on almost all indicators of macrophyte diversity (except for emerged vegetation). Similar relationships have been demonstrated for reservoirs of other origins. According to Mioduszewski [57], a lake catchment consisting of 40% wetlands can retain over 90% of pollution from agricultural sources. Wilk-Woźniak et al. [58] and Dudzińska et al. [59] found a positive correlation between macrophyte diversity and the proportion of forests in the catchment.

The effects of salinity on the ecosystem health of coastal lakes are even better known [60] than the impact of lake watersheds. It is commonly reported that salinity leads to a decline in biodiversity and disrupts macrophyte structure in coastal ecosystems, e.g., [29,61,62]. This may help to explain the fact that, in coastal lakes with low salinities < 7 PSU, the average values of diversity indices H’ and evenness J’ are lower than the values of the same macrophyte diversity indices for the trophic (harmonic) lakes that are typical of most lakes in Poland [28]. Based on macrophyte communities, the average value of H’ for the ten coastal lakes was 1.35 (including brackish lake H’ = 1.42; transitional lakes H’ = 1.20, freshwater lakes H’ = 1.45), while H’ for shallow inland trophic lakes was 1.41 (*n* = 78). The average value of J’ for coastal lakes was 0.45 (including brackish lake J’ = 0.68; transitional lakes J’ = 0.43, freshwater lakes J’ = 0.41), while J’ for shallow inland trophic lakes was 0.58 [27].

Among the lakes studied, Ptasi Raj stands out for its high salinity and macrophyte diversity, especially charophytes. The special conditions of Ptasi Raj result from its location on Sobieszewska Island within the Vistula river delta, which limits the catchment area and the external input of, for example, pollutants. About 62% of the lake’s catchment area (Figure 1) is forested (*Alnus glutinosa* plantings) and free of agricultural crops. Due to anthropogenic changes in the lake morphometry in the mid-19th century, resulting from the straightening and channelisation of the Vistula Channel (Wisła Śmiała) across sand dunes in the settlement of Górki near Gdańsk [63], lake Ptasi Raj gained a constant exchange of water with the sea through two culverts in a dyke. The input of saline water increases during periods of backwater from the sea. Noteworthy are the sandy soils in the watershed, which are typical of dune landscapes with a lack of hydrographic network (arheic area). The infiltrative nature of the initial soils in the south-eastern part of the watershed area, which are susceptible to wind erosion [6,10], causes the water of Ptasi Raj to have a relatively high salinity–the highest among the lakes studied in the southern Baltic coastal zone [62].

The specific conditions of Ptasi Raj, a brackish-type lake, affect the development of macrophyte species, both cosmopolitan species such as *P. australis* and charophytes-species typical of marine habitats, and found nowhere else in the lakes studied (Appendix A). According to Blindow et al. [64], increased salinity may affect both the sex and vegetative reproduction of charophytes, whose populations and diversity are declining in many regions of the world, including the brackish Baltic Sea [65]. Other authors, e.g., [66,67], have reported that changes in salinity can affect the growth rate of charophytes, their morphological characteristics, encrustation, etc.

In transitional lakes, a stronger influence of anthropogenic catchments of urban type on macrophyte diversity indicators such as the number of plant communities, H-diversity index, and share of submerged vegetation was observed. Similar relationships were observed in freshwater lakes as in inland lakes [28], where the proportion of submerged and emerged vegetation depended on the supply from agricultural and urbanised areas in their catchment. In view of this, it is preferable to increase the share of forests and urban greenery in the catchment area.

We share the opinion of Wells et al. [68] and Perez-Ruzafa et al. [69] that assessing the ecological status of coastal lakes can be difficult despite the wide variability in plant communities observed along environmental gradients (=salinity) of lakes at different stages of development. It is obvious that macrophytes alone cannot be used to assess the ecological status of a water body since the loss of vegetation cannot always be attributed to anthropogenic influences. Therefore, proper assessment of the ecological condition of a water body must be based on the integration of carefully selected biometric data appropriate to both specific external and internal conditions. Among many others, ecological assessment indices [19], depth limits of individual macrophyte species [20], and prediction models adapted to macrophytes [70] have been proposed for water quality assessment. In Estonia, several macrophyte metrics, such as the abundance of *Chara aspera*, *Ch. tomentosa*, *Utricularia vulgaris*, and *Cladium mariscus,* have been developed and validated to assess the ecological status of Estonian coastal lakes [2]. However, under the conditions of the southern Baltic coast, any metrics based on only *Chara* sp. would probably fail since only one coastal lake of the brackish type has been identified. This shows that any macrophyte community-based analysis of the ecological status of coastal lakes should be preceded by a site-specific survey.

It is worth noting that many species of aquatic macrophytes decontaminate organic and inorganic pollutants [71] and act as a natural, effective, and environmentally friendly technology (phytoremediation) that protects aquatic ecosystems from excessive pollutant loading. However, the spread of rush vegetation, especially reeds, together with the aeolian processes occurring near the coastal lakes or, in the case of Ptasi Raj, in their watersheds, contribute to their overgrowth and shallowing. In transitional and freshwater lakes, we observed an intensification of this process associated with a decrease in macrophyte diversity indicators (Figure 3, Figure 4 and Figure 5, Appendix A). Distinguishing the causes of bulrush expansion in transitional and freshwater lakes is difficult (Figure 5B). Both types of lakes appear to be associated with anthropogenic catchments, and the gradual filling of coastal lakes with the sands of shifting dunes and boulders intensifies. In addition, an important cause of the shallowing of transitional lakes and the development of rushes is the flow through the lakes (e.g., Łebsko), where large amounts of suspended sediment are deposited by rivers. The ability of *P. australis* to proliferate in a wide range of habitats, especially in areas with many physical disturbances, is well known [72]. Similarly, the cosmopolitan nature show *T. angustifolia* [73], *G. maxima* [74], or *Scirpus lacustris* [75].

We agree with Carvalho et al. [11] that the role of watersheds and their land uses have been underestimated and overlooked in water management programmes for coastal lakes. The lack of watershed management plans can lead to overuse, pollution, or physical alteration of coastal lakes. This can also lead to social conflicts among users of the limited but highly valuable coastal space [76]. Therefore, for a reliable assessment of the ecological status of coastal lakes as Natura 2000 habitats, the role of catchment size and its use/cover needs to be evaluated, which requires a revision of the WFD approach.

## 5. Conclusions

Our study is the first comprehensive attempt to investigate and evaluate the suitability of selected macrophyte metrics for land-use structure in direct and indirect coastal lake watersheds and for haline stress. The demonstrated relationships require the proper use of spatial planning tools. With this in mind, we propose to map land-use classes in coastal lake watersheds, defined by the CORINE Land Cover System (CLC), which is required for effective coastal lake monitoring and management. This will enable a rational, territorial organisation of land uses and their interconnections that balance development needs with the conservation needs of lakes to achieve social and economic goals consistent with the objectives of the Water Framework Directive. Our results contribute to the better application of macrophyte assessment methods that meet WFD requirements and to the better interpretation of coastal lake macrophyte diversity characteristics in the context of watershed land use.

We propose to apply the principles of integrated watershed management [77]. Unfortunately, it is difficult to formulate related policies at the national or regional level that are sufficiently detailed and practical to be implemented at the local level. Therefore, we emphasise the need to incorporate them into land-use planning, which is the responsibility of local authorities. This is particularly important given the various threats to coastal lakes; some of these, such as siltation, already exist, while others are very likely to appear, such as a sea-level rise or severe droughts, which will be accelerated by global climate change.

## Figures and Tables

**Figure 1 ijerph-19-16620-f001:**
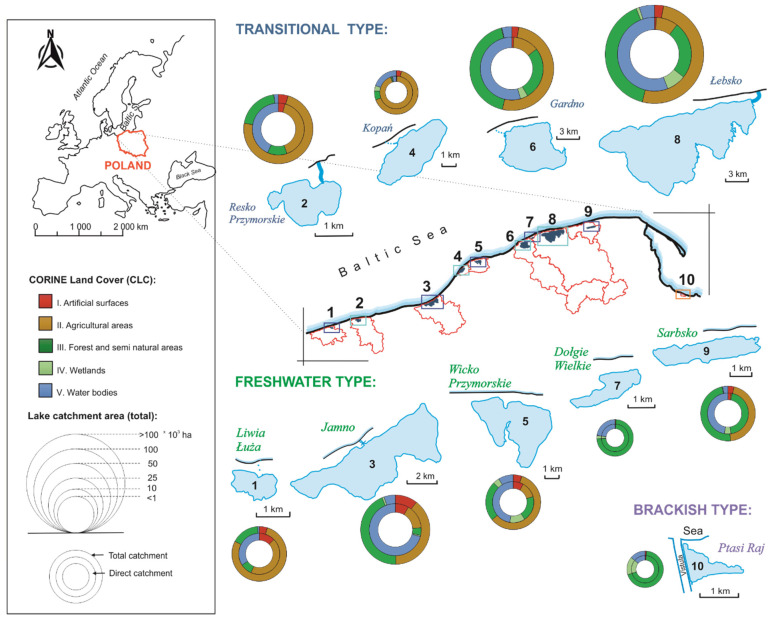
Location of studied coastal lakes with their watersheds (divides are marked by red lines) along the southern coast of the Baltic Sea, lake typology according to salinity level, as well as lake catchment areas and their land use (source: Copernicus Land Monitoring Service 2018 [23]).

**Figure 2 ijerph-19-16620-f002:**
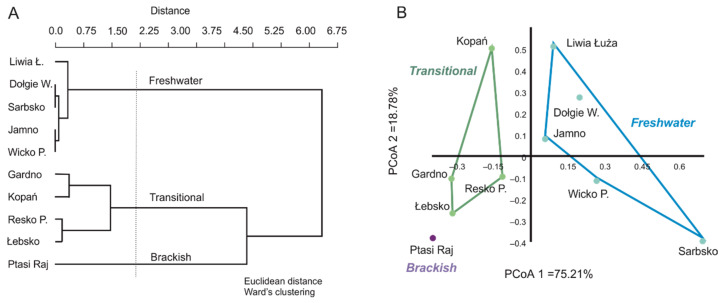
Procedure for the classification of the coastal lakes studied. (**A**) Classification of coastal lakes by salinity (PSU) using Euclidean distance and Ward’s agglomeration method. Sneath’s criterion was applied to determine clusters for lake types; (**B**) PCoA biplot showing three lake groups distinguished by salinity.

**Figure 3 ijerph-19-16620-f003:**
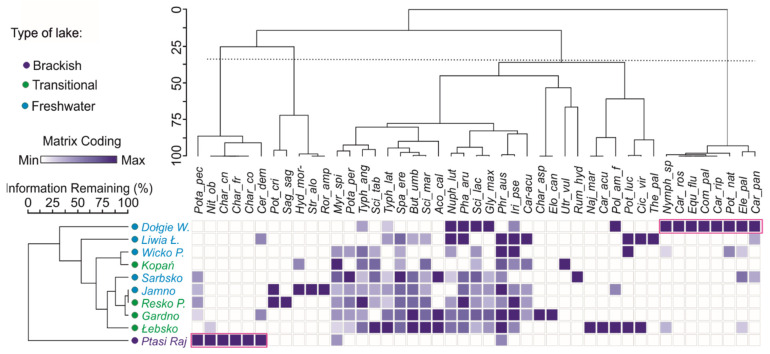
Two-way cluster analysis (TWCA) for macrophyte species in the studied coastal lakes. Colours depict values relative to column. The clusters combined into a single heat map to visualise associations between lakes (horizontal dendrogram) and macrophytes (vertical dendrogram). Heat map colours indicate minimum (white) to maximum (blue) contribution of each species in a lake. Red outlines indicate specific macrophyte clusters found in Ptasi Raj, a brackish lake, rich in charophytes, and in Dołgie Wielkie lake, with vegetation typical of humic acid-rich reservoirs.

**Figure 4 ijerph-19-16620-f004:**
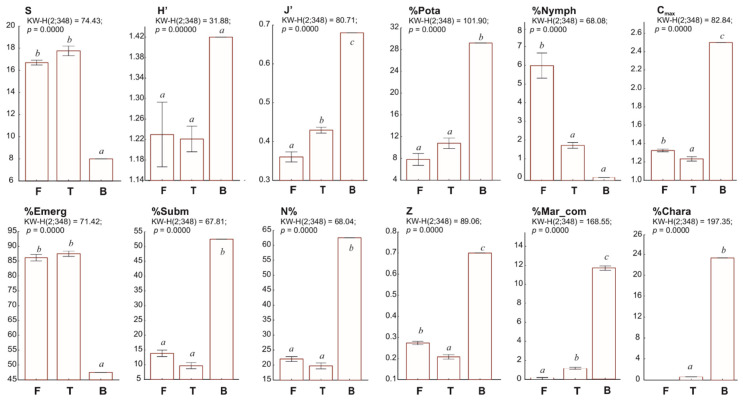
Composition and syntaxonomic diversity of macrophytes in the three types of Baltic coastal lakes: freshwater, transitional, and brackish. Standard error bars are given for all mean values. One-way ANOVA (degrees of freedom, F-statistic, and *p*-value) for each diversity metric are provided in the header of each graph. Statistically significant differences in means for lake groups according to the Tukey HSD multiple comparison test for 95% confidence interval are indicated by different letters (a, b, and c).

**Figure 5 ijerph-19-16620-f005:**
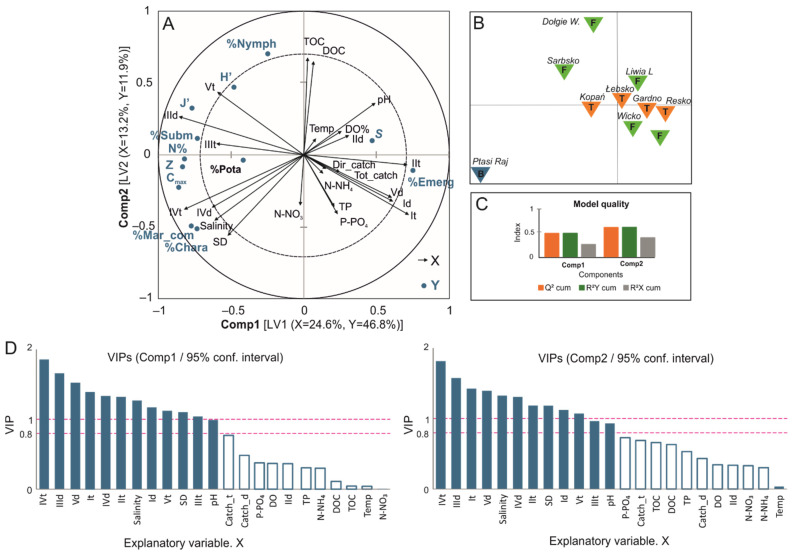
(**A**) Partial least squares (PLS) regression biplot reflecting the effect of land-use classes (CLC I–V), total and direct catchment areas, and water quality parameters as the explanatory variables (X) on the diversity of macrophytes (Y). The percentages of the variances in X and Y explained by each variable (as latent variables LV 1 and LV 2) are indicated on the respective axes. Inner dashed circle denotes correlation coefficient r = 0.75. (**B**) Biplot of lake ordination; (**C**) PLS-R model quality. (**D**) VIPs (Variable Influence on Projection) for each explanatory variable of LV 1 and LV2. VIP diagrams show relative importance of predictors. VIPs > 0.8, based on Wold’s criteria, indicate that the predictor variable is considered to be significantly important to the corresponding dependent variable. The addition of “t” to a land-use code indicates the entire watershed, and “d” the direct watershed.

**Table 1 ijerph-19-16620-t001:** Variable identification coefficients (VID ≥|0.85|) in the PLS models describing the effects of land use forms on macrophyte diversity indicators for freshwater (*n* = 5) and transitional (*n* = 4) types of lakes. Brackish lake represented by only one lake (Ptasi Raj) with domination of marine species was excluded from the list. Codes supported with subscript “d”–denote direct watershed and “t”–total watershed.

S		H’		J’		N%		C_max_		%Emerg		%Subm		Z		%Mar_comm	
CLC ^1^	VI_D_	CLC	VI_D_	CLC	VI_D_	CLC	VI_D_	CLC	VI_D_	CLC	VI_D_	CLC	VI_D_	CLC	VI_D_	CLC	VI_D_
Freshwater lake type																	
242_t_	0.91	313_d_	0.95	313_d_	0.99	313_d_	0.93	313_d_	0.99	313_d_	−0.92	313_d_	0.92	313_d_	0.98	112_t_	0.92
313_t_	0.88	324_t_	0.93	324_t_	0.94	324_t_	0.91	324_t_	0.95	324_t_	−0.91	324_t_	0.91	324_t_	0.94	112_d_	0.87
				312_d_	0.87			312_d_	0.86					312_d_	0.80	313_d_	−0.98
				331_d_	0.83			112_d_	−0.87							324_t_	−0.92
				324_d_	0.82			112_t_	−0.86							312_d_	−0.88
				112_t_	−0.90												
				112_d_	−0.87												
Transi_t_ional lake type																	
512_d_	0.99	121_t_	0.93	411_d_	0.99	313_d_	0.96	313_d_	0.99	324_t_	0.99	242_t_	0.95	313_d_	0.98	313_t_	0.99
243_t_	0.88	331_t_	0.93	313_d_	0.98	411_d_	0.81	411_d_	0.96	243_t_	0.97	411_t_	0.94	411_d_	0.87	121_t_	0.98
313_t_	0.88	412_t_	0.93	411_t_	0.90	324_t_	−0.85			411_t_	−0.99	112_d_	0.91	411_t_	0.85	331_d_	0.98
331_d_	0.87	411_d_	0.89	512_t_	0.86					242_t_	−0.97	231_d_	0.89			331_t_	0.98
324_t_	0.86	141_t_	0.88	242_t_	0.85					512_t_	−0.97	243_t_	−0.98			412_t_	0.98
311_t_	0.85	331_d_	0.88									324_t_	−0.98			141_t_	0.87
231_d_	−0.98	313_t_	0.88									512_d_	−0.92			211_t_	−0.99
112_d_	−0.99	211_t_	−0.91													211_d_	−0.94
112_t_	−0.92	211_d_	−0.86													112_t_	−0.85
211_t_	−0.85															112_d_	−0.85
																231_d_	−0.85

^1^ Corine Land Cover 2018 codes: 112: Discontinuous urban fabric; 121: Industrial or commercial units; 141: Green urban areas; 142: Sport and leisure facilities; 211: Non-irrigated arable land; 231: Pastures; 242: Complex cultivation patterns; 243: Land principally occupied by agriculture, with significant areas of natural vegetation; 311: Broad-leaved forest; 312: Coniferous forest; 313: Mixed forest; 321: Natural grasslands; 324: Transitional woodland-shrub; 331: Beaches, dunes, sands; 411: Inland marshes; 412: Peat bogs; 421: Saline marshes; 511: Water courses; 512: Water bodies. Marine communities in freshwater coastal lakes were well adapted to sparse residential built-up areas (CLC112), but they were quite unique in forested catchments (CLC 313_d_, CLC 324_t_, and CLC 312_d_), which did not promote the presence of rush phytocenoses *Scirpus* spp. The marine species in transitional lakes were also negatively affected by mixed forests (CLC 313_t_). The positive impact of industrial or commercial units (CLC 121_t_), beaches, dunes, sands (CLC 331_t_), and peat bogs (CLC 412_t_) in total catchments of coastal lakes was confirmed. The most limiting factor for marine species was non-irrigated arable lands (CLC 211_t_ and 211_d_) with VID = −0.99 and VID = −0.94, respectively. Less prominent were sparse urban areas CLC 112_t_ and CLC 112_d_ (VID = −0.85).

## Data Availability

Not applicable.

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
