# Peer review of "Response of Macrophyte Diversity in Coastal Lakes to Watershed Land Use and Salinity Gradient"

_ijerph, 2022, doi:10.3390/ijerph192416620_

Round 1

Reviewer 1 Report (Previous Reviewer 2)

Thank you for reconsidering my comments. I accept manuscript In prezent form

This manuscript is a resubmission of an earlier submission. The following is a list of the peer review reports and author responses from that submission.

Round 1

Reviewer 1 Report

Dear Authors,

this manuscript is a exeptional study written in very good style. It consists of valuable data about functions of bioindicators of macrophytes, but also usefull sugestions about WFD.

However, I recommend that the phrase community and phytocoenose should be avoided because the subject of theis manuscript are not typical phytocoenoses, due to the fact the study is not related to aquatic and marsh vegetation (e.g. ass. Nymphetum albo-luteae Nowinski 1928, ass. Ceratophylletum demersi (Soó 1927) Hild. 1956, ass. Scirpo-Phragmitetum W.Koch 1926, etc.).

Namely, authors precisely say: “Coastal vegetation was studied on selected transects up to 500 m wide, along which macrophyte communities were identified as clusters/aggregations of plants, mostly consisting of one species ≥1 m2, covering at least 25% of this area“.

Much more precise would be using aggregations or clusters or population of macrophytes insted of macrophyte communities/phytocoenoses.

In the next references, in the part References, years should be bolded: 14, 26, 27, 52, 63.

 In the Table A3 (in Supplementary file), correct Chara cnescens to Chara canescens.

In the Table A3 (in Supplementary file), correct Nympheides to Nymphaeides.

Sincerely,

Reviewer

Reviewer 2 Report

In these form - I do not accept the manuscript. take the following observations into account - I think the manuscript will be a good one.

1. Please provide better quality figures.

2. Authors provide many self-citations. It doesn’t look good. Eg. 29 and 30 in part of water sampling. Both articles provide standard description of water sampling. Self-citations are inappropriate. Please redefine this proceder.

3. In table A1 - provide information from where are the data!

4. Statistica™ 13.1 (Tebsco Inc. 2021). in line 246 - wrong. Should be Tibco Inc. 

5. material and methods: the authors present their discussions and reflections. Such a description should not be placed here. Additionally, in the description of the study site there is a description of the statistical methods used (line 156, 157) - this is also no place for such a description.

6. In paragraph 2.3 - does authors used standard methods for determination on water quality parameters?

7. In another paragraph 2.3 - databases and statistical analyses - please change number of paragraph. Please reconsider the name of paragraph. I see lack of database description nor information of repository of such database. 

8. In figures you use heat maps - please describe them in in the proper place.

9. Line 435 - 438 - not true. According to Fig. 5.

In my professional opinion - it’s enough to reject article in current form.

Reconsider statistical explanation of your studies.